# A scoping review of patient engagement activities during COVID-19: More consultation, less partnership

**Lauren Cadel**[1,2], **Michelle Marcinow**[1], **Jane Sandercock**[1], **Penny Dowedoff**[1], **Sara J. T. Guilcher**[2,3], **Alies Maybee**[4], **Susan Law**[1,3], **Kerry Kuluski**[1,3]*

**1** Institute for Better Health, Trillium Health Partners, Mississauga, Ontario, Canada, **2** Leslie Dan Faculty of Pharmacy, University of Toronto, Toronto, Ontario, Canada, **3** Institute of Health Policy, Management and Evaluation, University of Toronto, Toronto, Ontario, Canada, **4** Patient Partner, Canada

* kerry.kuluski@thp.ca

## Abstract

### Background

The COVID-19 pandemic has had a devastating impact on healthcare systems and care delivery, changing the context for patient and family engagement activities. Given the critical contribution of such activities in achieving health system quality goals, we undertook to address the question: *What is known about work that has been done on patient engagement activities during the pandemic*?

### Objective

To examine peer-reviewed and grey literature to identify the range of patient engagement activities, broadly defined (inclusive of engagement to support clinical care to partnerships in decision-making), occurring within health systems internationally during the first six months of the COVID-19 pandemic, as well as key barriers and facilitators for sustaining patient engagement activities during the pandemic.

### Methods

The following databases were searched: Medline, Embase and LitCOVID; a search for grey literature focused on the websites of professional organizations. Articles were required to be specific to COVID-19, describe patient engagement activities, involve a healthcare organization and be published from March 2020 to September 2020. Data were extracted and managed using Microsoft Excel. A content analysis of findings was conducted.

### Results

Twenty-nine articles were included. Few examples of more genuine partnership with patients were identified (such as co-design and organizational level decision making); most activities related to clinical level interactions (e.g. virtual consultations, remote appointments, family visits using technology and community outreach). Technology was leveraged in almost all reported studies to interact or connect with patients and families. Five main

**Data Availability Statement:** All relevant data are within the manuscript and its Supporting information files.

**Funding:** This paper was funded by the Canadian Foundation for Health Care Improvement. Dr. Kerry Kuluski holds the Dr. Mathias Gysler Research Chair in Patient and Family Centred Care at the Institute for Better Health, Trillium Health Partners. The funding for this Chair, supported by the Trillium Health Partners Foundation, was used to support Dr. Kuluski's time in leading the study reported in this paper.

**Competing interests:** The authors have declared that no competing interests exist.

descriptive categories were identified: (1) Engagement through Virtual Care; (2) Engagement through Other Technology; (3) Engagement for Service Improvements/ Recommendations; (4) Factors Impacting Patient Engagement; and (5) Lessons Learned though Patient Engagement.

## Conclusions

Evidence of how healthcare systems and organizations stayed connected to patients and families during the pandemic was identified; the majority of activities involved direct care consultations via technology. Since this review was conducted over the first six months of the pandemic, more work is needed to unpack the spectrum of patient engagement activities, including how they may evolve over time and to explore the barriers and facilitators for sustaining activities during major disruptions like pandemics.

## Introduction

The World Health Organization declared the COVID-19 outbreak, originating in Wuhan, China, as a pandemic on March 11, 2020 [1]. As of June 2021, over 174 million cases had been confirmed worldwide, with more than 3.6 million deaths [2]. Two hundred and nineteen countries/regions have been affected by this pandemic [2]. COVID-19 has not only affected healthcare systems, it has had a major impact on everyday life involving local and national economies [3], education systems [4], travel industries [5], immigration [6] and the environment [7]. In relation to healthcare systems, continuing to deliver high quality care amidst a pandemic has been challenging.

How healthcare systems interacted with patients and caregivers (forms of patient engagement) has changed over the course of the COVID-19 pandemic. The pandemic caused a major shift in health system activities with implications on how care was delivered and how patients and families were engaged as partners. For example, visitor restrictions were implemented in hospitals and long-term care facilities around the world [8–10], non-COVID-19 related research in some contexts was suspended [11] and many patient and family advisory committees, at least in the North American context, were put on hold [12]. An editorial by Richards and Scowcroft (2020) from the United Kingdom also described the abandonment of patient and public involvement in health system decisions and the omission of patient feedback on policies and guidelines [13].

Patient and family engagement in care and program planning is fundamental to person-centered care and quality of care [14, 15]. Over the past several decades, health systems have been working towards greater involvement of patients and families through engagement activities, such as the development of patient and caregiver advisory committees, consultations on guidelines or protocols, receiving feedback on experiences to promote change in care delivery and the healthcare system and working in partnership using co-design methods to design new programs and service delivery models [16–19]. Through patient engagement, care can be transformed by individuals who are directly impacted by healthcare decisions [20]. While these examples refer to more 'active' forms of engagement, activities can be more broad in nature, ranging—from clinical consultations to partnership activities (advisory groups and participation in policy decision making) [21]. Engagement, in these various forms, is intended to optimize care to meet the health and social care needs of patients and caregivers [22, 23].

Patient engagement can be defined as the partnership between patients, families and health professionals across different levels of the healthcare system with the goal of improving health and healthcare [21]. Reflecting on the broad range of engagement activities, Carman and colleagues' (2013) well cited, multidimensional framework for patient and family engagement outlines three core levels of engagement: 1) direct care, 2) organizational design and 3) governance and policy making. At each level, there is a continuum of engagement from consultation, to involvement, to partnership and shared leadership [21] reflecting both active and passive forms of engagement. Each stage of the continuum involves increased participation and collaboration from those being engaged, as they progress from information sharing to active partnership in the development and evaluation of healthcare programs and policies.

Patient engagement has been noted as an integral component of health care delivery, and is critical for obtaining feedback for improving processes, safety and experiences [24]. The pandemic halted many non-essential services; however, continuing to engage with patients and families during a pandemic is important to ensure quality of care is continually provided to those who are using and interacting with the healthcare system [14, 15]. With a rapidly changing environment as a result of COVID-19, understanding the initial impact of the pandemic on patient engagement activities requires our attention. The purpose of this scoping review was to examine peer reviewed and grey literature to identify the broad range of patient engagement activities that were happening within health systems during the first six months of the COVID-19 pandemic (March 2020 to September 2020), as well as key barriers and facilitators for patient engagement activities. We acknowledge that a scoping review is only one way to identify the type and extent of evidence available relating to patient engagement activities during the initial stages of COVID-19. Given that COVID-19 related papers were being published rapidly, our aim was to scope and synthesize the literature and to highlight knowledge gaps to help focus areas of improvement and future work [25].

## Materials and methods

This scoping review followed the methodological framework for scoping reviews outlined by Levac and colleagues [26], as well as the PRISMA-ScR reporting guidelines by Tricco and colleagues (see S2 Table) [27]. A protocol was developed and registered on Open Science Framework (https://osf.io/khuap).

### Identifying the research question

The following research question was formulated to guide this scoping review: *What is known in the literature about work that has been done internationally within healthcare organizations on patient engagement during the first six months of the COVID-19 pandemic*? Based on this question, the two main objectives were: (1) to identify strategies that have been implemented or sustained to engage with patients during the COVID-19 pandemic and (2) to identify the key barriers and facilitators to engaging with patients during the COVID-19 pandemic.

### Identifying relevant studies

A search strategy was developed with guidance from the Medical Library Association's resource on COVID-19 Literature Searches [28], as well as in consultation with a librarian at Trillium Health Partners (see S1 Table). The following are examples of terms searched using Boolean operators, truncations, wild cards and proximity operators: coronavirus, COVID-19, Sars-CoV-2, patient, family and caregiver engagement, participation, design and partner. Three databases were searched on May 25, 2020: Medline (Ovid Interface), Embase (Ovid Interface) and LitCOVID [29]. A search for relevant grey literature was conducted using

Google (Advanced Google searches), and the websites of key health system and guidance producing organizations and associations (e.g. World Health Organization, United Kingdom National Health Service). Due to the high volume of publications related to COVID-19, grey literature was searched regularly throughout this study until September 1, 2020 (when analysis and manuscript writing began).

## Study selection

For inclusion, articles were required to meet the following criteria: (1) specific to the context of COVID-19; (2) included a description of patient engagement activities; (3) involved a healthcare organization; and (4) published from March 11, 2020 to September 1, 2020. To be considered a patient engagement activity, we followed Carman's framework, which includes engaging with patients, families, their representatives, and health professionals in direct care (e.g. participating in clinical consultations, participating in support groups), organizational design and governance (e.g. serving on advisory councils, assisting with hiring committees) and/or policy making (e.g. collaborating with policy-makers, setting priorities for healthcare resources) [21]. The patient engagement activity did not have to be specific to COVID-19, just occurring during the pandemic. Based on the rapidly changing environment, this review was bounded within the first six months after the pandemic was declared in order to gain an understanding of the initial impact of COVID-19 on patient engagement activities. Articles were excluded if they were: (1) conference abstracts and articles in which the full-text was not available or accessible; or (2) articles published prior to 2019. Articles were excluded if the full-text was not accessible because information would not be able to be extracted from it. Articles published prior to 2019 were excluded because they would not be related to the COVID-19 pandemic.

The database searches identified 762 articles and the grey literature searches identified 11 articles (reports, commentaries, etc.; see Fig 1). Following deduplication using EndNote X9 [30], 473 articles remained for screening. Covidence, a software platform for systematic and scoping reviews [31], was used to facilitate the screening processes. The research team (KK, LC, MM) independently screened the titles and abstracts of 20 articles to test their agreement; with an agreement of 90%, the remaining articles were divided amongst the three reviewers and double screened (two individuals screened each article for inclusion). All disagreements were discussed in virtual meetings until consensus among the team was achieved. Articles that were included were then reviewed at the full-text level. A subset of full-text articles (n = 10) were independently screened by the research team (KK, LC, MM) to test their agreement (90%). All full-text articles were double screened and disagreements were resolved through virtual meetings.

## Charting the data

A data extraction form was developed in Microsoft Excel by the research team (KK, LC, MM). Data were extracted from the full-text articles by a single person (LC) to ensure consistency. A spot check of extracted data was performed (MM) for 10% of included articles to ensure the extracted data was complete and accurate. Extracted data contained the following information: study characteristics, population characteristics, patient engagement characteristics (description, target population, challenges, lessons learned, level of engagement [21]), intervention characteristics (if applicable), study outcomes and conclusions.

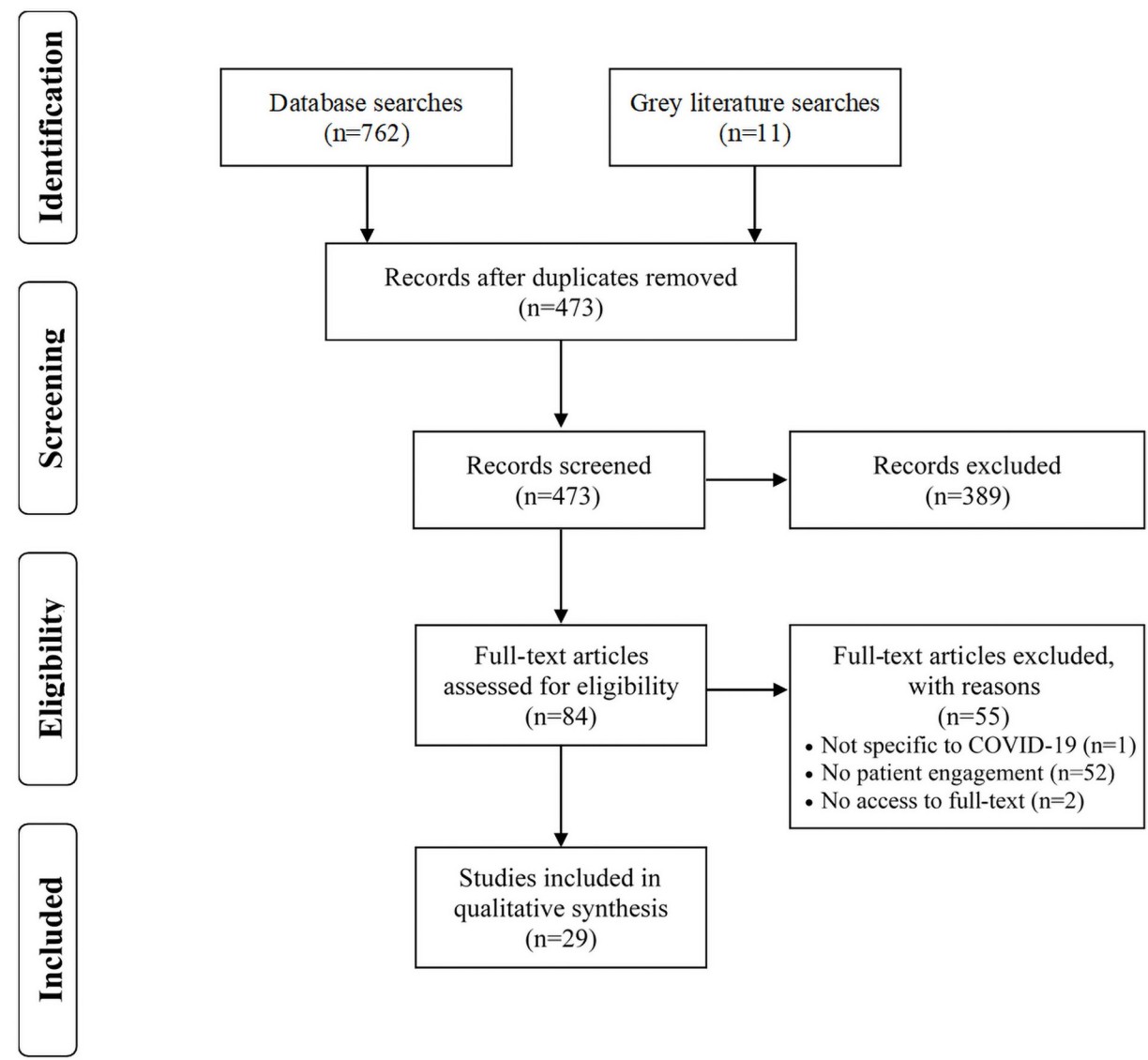

**Fig 1. PRISMA flow diagram of included articles.**

## Collating, summarizing and reporting the results

The extracted data were analyzed using descriptive quantitative and qualitative approaches. Descriptive quantitative analyses included summarizing the number of articles based on the method of data collection, year of publication, country in which it was conducted and the type of patent engagement activity. Carman and colleagues' framework was initially used to categorize each of the activities based on their level of engagement [21]; however, in doing so, it was identified that the majority of activities were categorized as direct care. A more in-depth analysis, guided by Hsieh and Shannon's conventional approach to content analysis [32], was then conducted by two authors (KK, LC) to identify the nuances and details of each activity. Both authors read the extracted descriptions and purpose of the engagement activities, referring back to the full-text as needed, and applied a one or two word description to the activity. The authors then worked together (through virtual meetings and in-depth conversations) to review

the simplified descriptions of the engagement activities and identified broader commonalities, which allowed for all articles to be classified into five categories (described in detail in the Results). This analysis provided a more nuanced set of categories within Carmen's categories of engagement.

## Results

### Study characteristics

A total of 473 unique articles were screened at the title and abstract level and 83 were screened at the full-text level. Twenty-nine articles met the eligibility criteria and were included in this scoping review (see Fig 1). The majority of included articles were not original research articles (e.g. editorials, reviews, commentaries; n = 15). The most common study design of the empirical articles was quantitative (n = 12), with one mixed method study and one qualitative study. The work was carried out across 14 different countries, but the United States was the most common (n = 17). All of the included articles were published in 2020, with the majority published in May (n = 15). Table 1 outlines the characteristics of included articles.

During the analysis, five main descriptive categories were identified related to the literature on engagement: (1) Engagement through Virtual Care; (2) Engagement through Other Technology; (3) Engagement for Service Improvements/ Recommendations; (4) Factors Impacting Patient Engagement; and (5) Lessons Learned through Patient Engagement (see Fig 2). These categories are not mutually exclusive as some activities fell into more than one category. Across all categories, the majority of activities involved individual level patient engagement (i.e. information sharing and dissemination, feedback on attitudes and experiences), with limited activities empowering patients as leaders, integrating patient and caregiver partners as team members or co-designing interventions, programs or initiatives with patients and caregivers. Despite not limiting the topic of patient engagement activities to COVID-19, we identified that most of the articles included activities specific to the pandemic. Table 2 describes the patient engagement characteristics.

### Engagement through virtual care

The primary type of patient engagement was through virtual care, with two sub-categories of engagement activities: (a) virtual care for health and/or social services (n = 13) [35, 37, 41, 43, 45, 46, 48, 51–54, 56, 58] and (b) virtual care for connecting with families and caregivers, including their involvement in care team discussions (n = 3) [35, 48, 49].

**Virtual care for health and social services.** Virtual care was primarily used for clinical services, inclusive of both health and social care (e.g. consultations, care conferences, home monitoring, remote appointments/check-ups) [35, 37, 41, 43, 45, 46, 48, 51–54, 56, 58]. Virtual visits and consultations extended beyond medical care to include social services [35, 37, 43, 48]. For example, healthcare providers and care coordinators at a HIV clinic in the United States used telemedicine to consult with patients, check on their medication supply, help set up technology (answer questions) and ensure food and housing security during COVID-19 [37].

In light of the COVID-19 pandemic, many healthcare organizations and institutions transitioned from in-person appointments to online platforms (i.e. telemedicine/ telehealth). This transition and increased use of telemedicine to encourage virtual interactions between healthcare providers and patients was described in several included articles [41, 46, 56, 58]. For example, from March 2020 to April 2020, telemedicine visits (telephone and video) at one institution increased from fewer than 100 visits per day to greater than 2,200 visits per day [56]. A number of platforms were used for providing virtual clinical services, with one article

**Table 1. Characteristics of included articles (n = 29).**

| Author (Year) | Country | Objective | Method | Study Design | Participants/ Target Population | Sample Size |
|---|---|---|---|---|---|---|
| Meinert et al. (2020) [33] | United Kingdom, France, and Sweden | • To describe the design and implementation of a digital health solution for older adults (Activating Digital to Support Social Distancing COVID-19 Aware Family Engagement (ADAPT-CAFÉ)) | Mixed Methods | Case study and feasibility study | Older adults, family members and peer groups | 27,450 projected users |
| Sirotich et al. (2020) [34] | United States | • To describe the development of a survey through rapid engagement with patients, patient organizations and rheumatologists<br>• To explore how COVID-19 is affecting the physical and mental health of people with rheumatic diseases | Quantitative | Cross-sectional study | Adults and parents of children with rheumatic diseases | 9,541 survey responses |
| Overton et al. (2020) [35] | United States | • To describe how the patient experience team is supporting programs and patients during the pandemic | Neither | Summary/ review | Patients and staff at MD Anderson | N/A |
| Ekzayez et al. (2020) [36] | Syria | • NR | Neither | Review/ perspective paper | NR | N/A |
| Mgbako et al. (2020) [37] | United States | • To describe patient care at an HIV clinic during COVID-19<br>• To identify how telemedicine may affect patient-centred care | Neither | Review/ notes from the field | Adults with HIV | 1 |
| Wei et al. (2020) [38] | China | • To develop an internet-based intervention for COVID-19 patients experiencing psychological distress<br>• To test the efficacy of the intervention on COVID-19 patients with depression and anxiety | Quantitative | Prospective, randomized, controlled, 2-week study | Laboratory confirmed COVID-19 patients in the isolation ward | 26 |
| Amir-Behghadami et al. (2020) [39] | Iran | • To describe the implementation of an electronic self-screening system for COVID-19 | Neither | Letter to the editor | Iranian residents/ patients with COVID-19 | N/A |
| Hart et al. (2020) [40] | United States | • To describe a framework for family-centred care in inpatient settings during COVID-19 | Neither | Review | NR | N/A |
| Tenforde et al. (2020) [41] | United States | • To present the process for the rapid adoption of telemedicine during COVID-19, as well as the results of a quality improvement initiative | Quantitative | Quality improvement initiative | Outpatient sports & musculoskeletal medicine patients and physicians | 132 |
| Hu and Qiu (2020) [42] | China | • To outline measures to improve infection prevention through the implementation of risk communication and community engagement | Neither | Review/ perspective paper | NR | N/A |
| Medina et al. (2020) [43] | United States | • To describe the effectiveness of a home monitoring program during COVID-19 | Quantitative | Intervention study | COVID-19 Patients | 1,924 |
| Griffin et al. (2020) [44] | United States | • To describe the rapid development and implementation of processes to prepare for the impact of COVID-19 | Neither | Review/ critical care perspective | COVID-19 Patients | N/A |
| Japan ECMOnet for COVID-19 (2020) [45] | Japan | • To describe a telephone consultation working group established for patients with severe respiratory failure caused by COVID-19 | Neither | Letter to the editor | COVID-19 Patients | 12 |
| Murphy (2020) [46] | United Kingdom | • To review diabetes management in pregnant women before and during the lockdown caused by COVID-19 | Neither | Description/ review of processes | Pregnant women with diabetes | N/A |

*(Continued)*

**Table 1.** (Continued)

| Author (Year) | Country | Objective | Method | Study Design | Participants/ Target Population | Sample Size |
|---|---|---|---|---|---|---|
| Yassa et al. (2020) [47] | Turkey | • To identify the attitudes, concerns and knowledge of COVID-19 among non-infected pregnant women | Quantitative | Cross-sectional survey research | Healthy, pregnant women over 30th gestational week | 172 |
| | | • To develop targeted messages and counselling based on their attitudes, concerns and knowledge | | | | |
| Kreimer (2020) [48] | United States | • To describe the changes made to palliative care neurologists' practice to address the fears and isolation of their patients and families | Neither | Article in brief | Physicians, neurologists | N/A |
| Mercadante et al. (2020) [49] | Italy | • To describe the use of WhatsApp to involve family members in clinical rounds and explore their perspectives | Qualitative | Descriptive | Family members of palliative patients | 16 |
| Padala et al. (2020) [50] | United States | • To explore the perspectives of older patients and their caregivers regarding clinical research during COVID-19 | Quantitative | Cross-sectional study | Older participants and caregivers enrolled in clinical studies | 51 |
| Peahl et al. (2020) [51] | United States | • To describe new guidelines and experiences transitioning to a new model for prenatal care | Neither | Description of guidelines | Low risk pregnant women | N/A |
| Fortune et al. (2020) [52] | Canada, United States, Europe, Mexico and Australia | • To describe the effects of a group videoconference-based intervention on anxiety, depression, stress, loneliness, boredom, physical activity, and frequency of social interactions | Quantitative | Partially nested randomised controlled trial | At-risk scleroderma patients | NR |
| Patel et al. (2020) [53] | United States | • To describe a pathway for increasing capacity for remote enrollment in telehealth | Quantitative | Report | Pediatric patients | Weekly enrollment: 0–12: 1582 13–17: 527 |
| Annis et al. (2020) [54] | United States | • To describe experiences with a remote patient monitoring program | Quantitative | Case report | COVID-19 Patients | 1,496 |
| | | • To investigate satisfaction and program acceptability among patients and staff | | | | |
| Edvardsson et al. (2020) [55] | Australia | • To describe the relational aspects of person-centred care in the context of research, practice and COVID-19 | Neither | Review/ contemporary issue | Patients during COVID-19 | N/A |
| Contreras et al. (2020) [56] | United States | • To describe the telemedicine environment, changes made because of COVID-19 and implications for the future | Neither | Review | NR | N/A |
| Brown et al. (2020) [57] | United States | • To examine the pain, anxiety, physical function and economic ability to have surgery among hip and knee arthroplasty patients whose surgery was delayed because of COVID-19 | Quantitative | Cross-sectional Survey/ questionnaire | Electively scheduled hip and knee arthroplasty patients | 360 |
| Peters and Garg (2020) [58] | United States | • To describe the experiences of two patients who engaged in telehealth relating to diabetes care | Quantitative | Case study | Adult patients with type 1 diabetes | 2 |
| Kullar et al. (2020) [59] | United States | • To review the use of Twitter for providing information about infectious diseases | Neither | Review | NR | N/A |
| Chen et al. (2020) [60] | China | • To explore how social media promoted citizen engagement during COVID-19 by the Chinese government | Quantitative | Pioneering study | NR | NR |
| Dokken and Ahmann (2020) [61] | United States | • To describe the development of, and the guidelines for, person and family-centred care during difficult times | Neither | Editorial | Family members of hospitalized patients | N/A |

Abbreviations: NR = not reported; N/A = not applicable; HIV = human immunodeficiency virus.

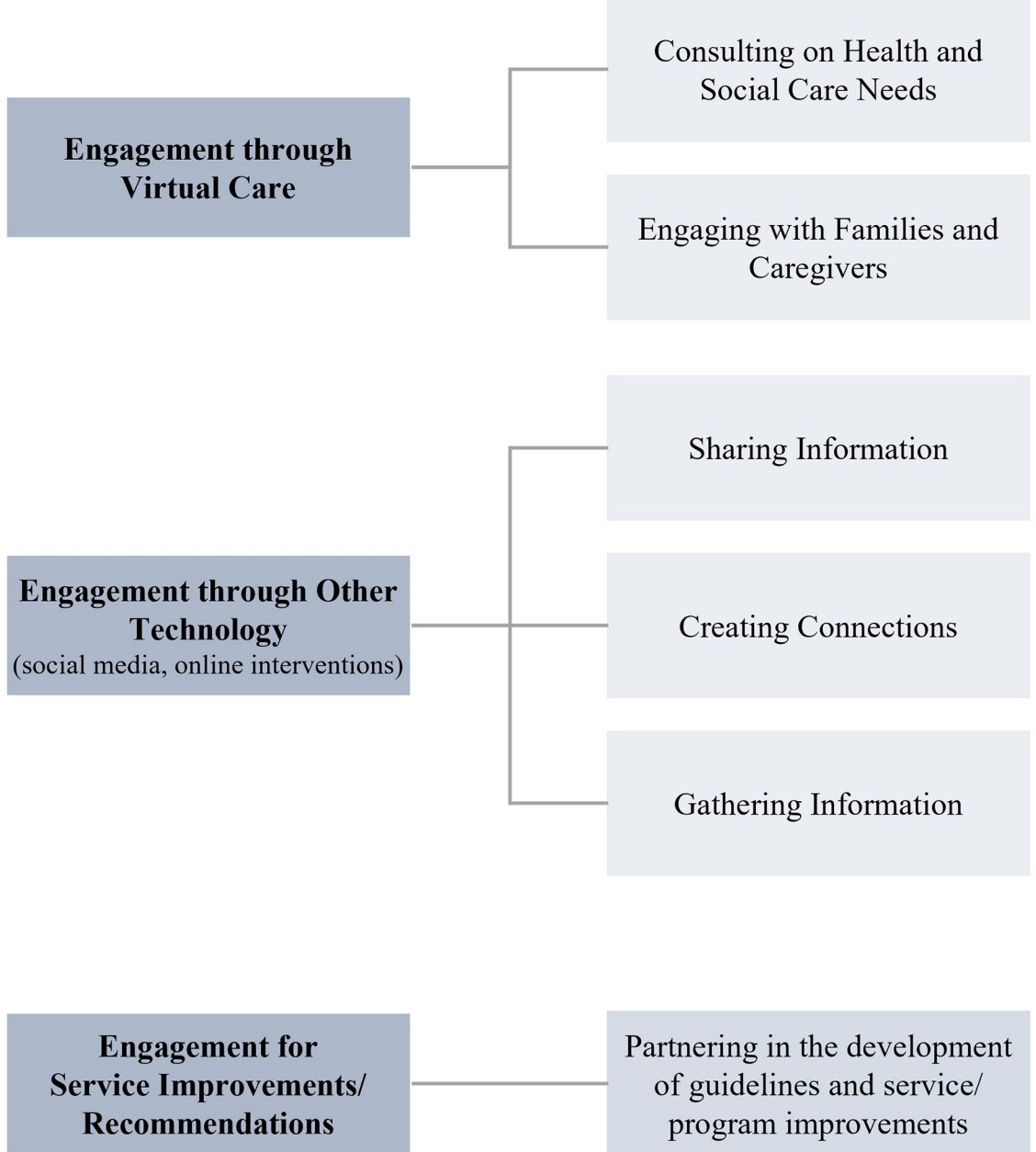

**Fig 2. Methods used by health systems to stay connected with patients and families during the COVID-19 pandemic.**

describing appropriate platforms on which to engage with patients (e.g. Skype for Business/ Microsoft Teams, Updox, Zoom for Healthcare, Doxy.me, Google G Suite Hangouts Meet, Cisco Webex Meetings/Webex Teams, Amazon Chime, GoToMeeting, Spruce Health Care Messenger) [56]. Further, authors reflected on the potential benefits of telemedicine for patients, including: the ability to reach more patients (i.e. rural, international, low-socioeconomic status) and improved access (eliminating parking, public transit, etc.) [37, 51, 56, 58].

Feedback on virtual care services was also obtained from patients and healthcare providers (examples combining virtual care for clinical services and technology for gathering

**Table 2. Characteristics of patient engagement activities.**

| Author (Year) | Engagement Activity | Description of engagement activity | Challenges with engagement | Lessons learned from engagement | Engagement Category or Sub-Category |
|---|---|---|---|---|---|
| Mgbako et al. (2020) [37] | Virtual Visits/ Consultations | • The use of telemedicine to connect with and check in on patients regarding their medication supply, food and housing situation and to help set-up technology | • Technological issues– connectivity disruptions<br>• Difficulties building provider-patient rapport via telemedicine | • Building rapport is critical for recruiting and engaging with marginalized populations and persons from hard-to-reach communities<br>• A team-based care model needs to be built into telemedicine | Virtual Care for Health and Social Services |
| Medina et al. (2020) [43] | Home monitoring program | • A program to provide information to patients on COVID-19, home isolation, social support, home safety and a care companion application (MyChart) | NR | • Integrating disciplines (psychiatry, behavioural health, social work, community partners) in the home monitoring program | Virtual Care for Health and Social Services |
| Japan ECMOnet for COVID-19 (2020) [45] | Telephone consultations | • A 24 hour/day service providing telephone consultations for patients with severe respiratory failure | NR | NR | Virtual Care for Health and Social Services |
| Murphy (2020) [46] | Remote engagement/ services | • The use of online resources (virtual training, email, telephone) to support pregnant women with type one diabetes with blood pressure and blood glucose monitoring | NR | NR | Virtual Care for Health and Social Services |
| Peahl et al. (2020) [51] | Virtual visits | • The use of telephone or video technology as touchpoints between in-person visits for addressing questions and completing depression screens | • Disadvantaged populations (patients living in rural areas, low socioeconomic status) | NR | Virtual Care for Health and Social Services |
| Fortune et al. (2020) [52] | Scleroderma Patient-centered Intervention Network COVID-19 Home-isolation Activities Together Program | • A video conference-based intervention targeting anxiety and mental health outcomes<br>• The patient advisory team helped design the program and trial | NR | NR | Virtual Care for Health and Social Services |
| Patel et al. (2020) [53] | Virtual enrolment in My Health at Vanderbilt and Telehealth Visits | • The use of technology to enroll pediatric patients in telehealth visits | • Teenagers who are unable to complete forms due to limited decision-making capacity or speech/ language barriers | • Using successful strategies from the COVID-19 response in future situations/ disasters | Virtual Care for Health and Social Services |
| Peters and Garg (2020) [58] | Telehealth | • Providing remote diabetes care using technology | • Technological barriers (patients don't have continuous glucose monitors, unsure how to find glucose levels with monitor, no availability of data)<br>• Cost of devices for underserved patients | • Changes in service delivery can improve access and outcomes for patients with diabetes | Virtual Care for Health and Social Services |

*(Continued)*

**Table 2.** (Continued)

| Author (Year) | Engagement Activity | Description of engagement activity | Challenges with engagement | Lessons learned from engagement | Engagement Category or Sub-Category |
|---|---|---|---|---|---|
| Contreras et al. (2020) [56] | Telemedicine | • Providing care in real-time through technology supporting audio-visual interactions | • Technological issues (bandwidth) and logistical difficulties (selecting a platform) | • Educating all stakeholders involved on implementation and use<br>• Distribution of tip sheets and instructional modules<br>• Naming clinical champions during rollout periods<br>• Considering comfort level and health literacy when using telehealth | Virtual Care for Health and Social Services |
| Overton et al. (2020) [35] | Virtual Care | • Use of MyChart (combination of Epic and Zoom) for video visits with patients<br>• Virtual care conferences (video or telephone) with caregivers and family members | NR | NR | Virtual Care for Health and Social Services Virtual Care for Connecting with Families and Caregivers |
| Kreimer (2020) [48] | Telehealth | • The use of telephone and video appointments for advance care planning and palliative patients<br>• Assisting patients with setting up Zoom and FaceTime to talk with family/friends | • Inability to see body language or read nonverbal cues<br>• Delivering bad news via technology | NR | Virtual Care for Health and Social Services Virtual Care for Connecting with Families and Caregivers |
| Tenforde et al. (2020) [41] | Telemedicine and follow-up survey | • The use of telemedicine (InTouch, Zoom) to facilitate communication between physicians and patients<br>• Completion of an online quality improvement survey following the visit | • Patients with cognitive issues, who are deaf, blind or hard of hearing, with limited technology literacy, do not have the required technology<br>• Patients who require an interpreter | • Logistic support for the physiatrist prior to telemedicine visits<br>• Education and training for physiatrists so they are comfortable with telemedicine<br>• Information provided to patients to complete the visit in a private location | Virtual Care for Health and Social Services Technology for Gathering Information |
| Annis et al. (2020) [54] | Remote Patient Monitoring and Patient Satisfaction | • A monitoring program specific to COVID-19 that provided patients with information about COVID-19, symptom assessments and reminders about hygiene and social distancing<br>• A survey on patient satisfaction with the program was administered following the completion of treatment | • Having enough staff to match to newly enrolled patients and respond to messages | • Improving the system to allow for mass messaging to patients and tools to measure patients coming into and out of the program<br>• Improving the process of enrolling patients | Virtual Care for Health and Social Services Technology for Gathering Information |
| Mercadante et al. (2020) [49] | WhatsApp for clinical rounds | • The use of WhatsApp (mobile messaging app) to involve family members in clinical rounds during visitor restrictions | • No access to a smartphone | • Ensuring the availability of/ access to technology | Virtual Care for Connecting with Families and Caregivers |

(*Continued*)

**Table 2.** (Continued)

| Author (Year) | Engagement Activity | Description of engagement activity | Challenges with engagement | Lessons learned from engagement | Engagement Category or Sub-Category |
|---|---|---|---|---|---|
| Ekzayez et al. (2020) [36] | Communication and information dissemination | • A volunteer campaign aiming to raise awareness, share information, identify high-risk individuals and connect community members to campaign resources<br>• Social media (WhatsApp and Facebook) was used to stay connected and share information | NR | NR | Technology for Sharing Information |
| Hu and Qiu (2020) [42] | Information sharing | • The use of online-based channels to share information about the pandemic and to respond to the publics' concerns | NR | • Sharing timely and accessible information through trusted sources<br>• Combatting rumors and misinformation with transparent information<br>• Sharing information internationally | Technology for Sharing Information |
| Chen et al. (2020) [60] | Sina Weibo account 'Healthy China' | • The use of Sina Weibo (social media platform) for communicating information and engaging with its followers | NR | NR | Technology for Sharing Information |
| Kullar et al. (2020) [59] | Twitter | • The use of Twitter (social media platform) as a tool for communicating with providers and the public to share information about infectious diseases | NR | NR | Technology for Sharing Information Technology for Creating Connections |
| Meinert et al. (2020) [33] | 1. Digital Health Solution<br>2. User experience (qualitative and quantitative)<br>3. User feedback and integration | 1. An app for older adults to maintain social interactions during physical distancing<br>2. Qualitative and quantitative feedback on user experience through in-app surveys, interviews and virtual focus groups<br>3. Virtual interviews with participants to receive feedback on the app to integrate into future iterations | NR | • Learning how to make the best use of resources, people and ideas<br>• Identifying what has worked for service delivery and embedding them in future care | Technology for Creating Connections |
| Edvardsson et al. (2020) [55] | Letters and technology for socialization | • Receiving letters from children to remain engaged with the community<br>• Using technology (email, social media, etc.) to maintain social relationships | NR | • Differentiate positive and negative outcomes and experiences<br>• Instrumentation should be sensitive to the life-stage of the targeted individual | Technology for Creating Connections |
| Griffin et al. (2020) [44] | Communicating with families | • The use of video conferencing to connect the care team or patients with family members | • Fragmented communication with families | NR | Technology for Creating Connections |

(*Continued*)

**Table 2.** (Continued)

| Author (Year) | Engagement Activity | Description of engagement activity | Challenges with engagement | Lessons learned from engagement | Engagement Category or Sub-Category |
|---|---|---|---|---|---|
| Sirotich et al. (2020) [34] | Survey development and dissemination | • Development and distribution of a COVID-19 Patient Experience survey in consultation with a steering committee, patient partners and patient organization representatives | NR | NR | Technology for Gathering Information |
| Wei et al. (2020) [38] | Self-help Intervention | • An internet-based, self-help intervention that integrates training on breath relaxation, mindfulness, refuge skills and the butterfly hug method<br>• The impact on depression and anxiety was measured | NR | NR | Technology for Gathering Information |
| Brown et al. (2020) [57] | Survey | • An electronic or telephone questionnaire about anxiety around COVID-19 and canceled operations, the patients' disease state and socioeconomic concerns | • Limited number of patient responses | NR | Technology for Gathering Information |
| Amir-Behghadami et al. (2020) [39] | Self-screening Tool | • A national, electronic, self-screening tool for Iranian residents to log information and complete questions about COVID-19 symptoms and other chronic diseases | NR | • Public education about the consequences of COVID-19 can contribute to the involvement in self-screening | Technology for Gathering Information |
| Hart et al. (2020) [40] | Framework for supporting family-centred care | • A framework of strategies for maintaining family-centred care when physical distancing measures are in place | • Family unavailable during daytime hours, no access to internet or devices with videoconferencing capability, lack of technology literacy, inability to speak the same language as the clinical team, lack of communication aids (glasses, hearing aids) | NR | Engagement for Service Improvements / Recommendations |
| Yassa et al. (2020) [47] | Survey | • A survey to understand pregnant women's knowledge, attitudes and concerns towards COVID-19, with the goal of developing targeted messages | • Illiteracy, challenges with translation | NR | Engagement for Service Improvements / Recommendations |
| Padala et al. (2020) [50] | Survey | • A survey to identify participant perspectives on safety, panic among the public and how the medical centre is handing the pandemic | NR | NR | Engagement for Service Improvements / Recommendations |
| Dokken and Ahmann (2020) [61] | Person-Centered Guidelines for Preserving Family Presence in Challenging Times | • The virtual development of guidelines for maintaining person and family centred care during difficult times (i.e. a pandemic) | NR | NR | Engagement for Service Improvements / Recommendations |

Abbreviations: NR = not reported; HIV = human immunodeficiency virus

information [41, 54]. For example, a quality improvement initiative received feedback from patients (n = 119) and physicians (n = 14) on the rapid implementation of telemedicine (via InTouch or Zoom) on a physical medicine and rehabilitation department in the United States [41]. Patients were engaged through telemedicine and follow-up surveys to identify their demographic characteristics (age, gender, insurance status, travel time saved), telemedicine details and characteristics (accompanied by family/friend, type, reason and time of visit) overall experiences (developing a treatment plan, communication, convenience, discussing concerns/questions, satisfaction) and future value. The majority of patients (>90%) rated their telemedicine experience as good or very good across the experience measures and most believed it would be valuable to complete a future telemedicine visit.

**Virtual care for connecting with families and caregivers.** Virtual care was also used to connect healthcare providers or patients with family members and caregivers during healthcare interactions [35, 48, 49]. WhatsApp, Zoom, FaceTime and MyChart (online tool combining Epic and Zoom) were the main platforms for connecting. Despite visitor restrictions, these platforms gave family members the opportunity to be involved in clinical rounds [35, 49] and remain socially connected with patients while they were in hospital [48]. One study explored family members' of palliative patients (n = 16) thoughts and experiences with the use of WhatsApp for sharing information on the patients' progress [49]. Overall, family members were happy to virtually attend clinical rounds, receive and exchange information during the call and see their loved one. However, participants also noted that the virtual connection did not replace their physical presence. Technology-based solutions allowed for improved communication and connection between healthcare providers and patients or family members, but the importance of providing clear communication about COVID-19 and self-isolation, addressing patients concerns' and demonstrating compassion throughout the pandemic were also highlighted [35].

## Engagement through other technology

Patient engagement activities through the use of other technology (e.g. social media, online-based interventions) were identified as a way of sharing information (n = 4) [36, 42, 59, 60], creating connections (n = 4) [33, 44, 55, 59] and gathering information (n = 6) [34, 38, 39, 41, 54, 57].

**Technology for sharing information.** The use of technology for engaging patients and the community through information sharing was identified in four articles [36, 42, 59, 60]. Information was shared through online platforms, including social media (e.g. Twitter, Sina Weibo). Online-based channels were used in China to share information about the pandemic, improve risk communication and community engagement practices, and respond to the publics' concerns [42]. The importance of sharing information through the communities' trusted sources, combatting rumors early by disclosing up-to-date information and ensuring information was accessible and comprehensible to the public was noted by the authors. Twitter was also used as a communication tool for healthcare providers and the general public to share information about infectious diseases [59]. Twitter chats and hashtags were discussed as valuable tools for both engaging with, and disseminating information to, other professionals and the public. During COVID-19, the authors explained the importance of Twitter for remaining up-to-date on the state of the pandemic and relevant literature being published, while ensuring credible resources were followed (@WHO, @CDCgov, @IDSAInfo). A combination of social media platforms (e.g. Facebook, WhatsApp) was also used by a volunteer campaign in Syria, Volunteers Against Corona, to effectively communicate, stay connected and share information about COVID-19 [36].

**Technology for creating connections.** Patient engagement activities using technology (digital health solutions, email, teleconferencing) for creating or maintaining social connections were identified in four articles [33, 44, 55, 59]. The importance of maintaining person-centred care, which included promoting communication between patients, families and healthcare providers and limiting relationship and social restrictions during the pandemic was discussed in two articles [44, 55]. Residents at a rural healthcare facility in Australia remained socially engaged with their family and friends through the use of email and social media [55].

Another paper described a future mixed methods case study which aims to develop an app for older adults, their families and peers to enhance their overall health and well-being during social distancing measures due to COVID-19 [33]. This digital health solution, Activating Digital to Support Social Distancing COVID-19 Aware Family Engagement (ADAPT-CAFÉ), will allow families and peers to remain in contact with older adults (in the community or in hospital) through virtual interactions. The app will also integrate goal setting, promote good nutrition and physical activity and track symptoms. During beta phase testing of the app, qualitative interviews will be conducted with participants and their feedback will be integrated in future iterations of the app, reflecting an example of co-design.

**Technology for gathering information.** Patient engagement activities using technology for gathering information (experiences, satisfaction, knowledge, attitudes, concerns, health outcomes) were identified in six articles [34, 38, 39, 41, 54, 57]. These activities included survey development and/or completion, self-screening and a self-help intervention.

The use of surveys to gain information and insights about participants' knowledge, attitudes, concerns, health outcomes and overall experiences (with their condition, care or engagement in activities) was identified in a number of articles [34, 38, 41, 54, 57]. These articles involved patients as partners in research activities (responding to surveys and participating in randomized controlled trials). There were few examples of activities that involved patients in multiple aspects of a research project; however an example of greater involvement was a global patient experience survey for individuals with rheumatic disease that was launched to identify patients' concerns with the disease and treatment and the impact of COVID-19 on their physical and mental health [34]. This survey was developed through rapid engagement of patients with rheumatic disease, patient organizations and rheumatologists. Patients were involved in all stages of the research, from study design to dissemination of the survey (i.e. development of research questions, study design, measuring instruments; participating in recruitment and the steering committee; and writing of the manuscript). Their main responsibilities included providing input on the content of the survey questions, reviewing the survey questions for culturally appropriate language and translating the survey into different languages. Involvement was facilitated through the use of an online messaging and collaboration tool. Sirotich and colleagues described a process for, and the benefits of, the rapid engagement of multiple stakeholders in order to address a complex problem.

## Engagement for service improvements/recommendations

Few articles (n = 4) involved patients as partners as part of the development of guidelines or service improvements [40, 47, 50, 61]. Two of the included articles conducted surveys to inform the design of services [47, 50]. A cross-sectional study was conducted by Yassa and colleagues (2020) in Turkey to understand pregnant women's knowledge, attitudes and concerns towards COVID-19, with the goal of developing targeted counselling and messages during the pandemic [47]. Based on the findings from the survey, the authors noted the importance of providing education, mental health support and counseling to pregnant women during the

COVID-19 pandemic; however, no specific programs or services had been designed at the time of our scoping review.

Additionally, two articles developed frameworks to ensure family support/ family-centred care during the pandemic [40, 61]. The "Person-Centred Guidelines for Preserving Family Presence in Challenging Times" (pg 1) [61] was created in virtual consultation with over 60 organizations, inclusive of patients, caregivers, advocates, clinicians and policy-makers. The guidelines were developed for healthcare leaders and health authorities and are intended to be applied across different contexts, including resource-challenged settings and among vulnerable populations. The goal of the guidelines is to balance individual needs with safety and community needs, support the principles of person-centred care across the continuum, keep patients and families connected through continual assessments, minimize risks and isolation, communicate expectations and policies clearly, support social connections, educate patients and families on discharge processes and partner with families. Some specific examples include keeping a digital diary of the patients' ICU experience (to help fill in the gaps for patients and families about what happened during hospital stay and to minimize stress when discharged home) and redeploying staff, volunteers or medical students to act as "connectors" between patients and families, who might be separated (due to hospital stay), particularly if there is no technology available.

### Factors impacting patient engagement

Just under half of the included articles (n = 13) described technological barriers or individual patient characteristics that impacted organizations' ability to engage with patients and families [37, 40, 41, 44, 47–49, 51, 53, 54, 56–58]. The most common technological barriers to engaging with patients were: technology that did not support video conferencing, limited technology literacy among those engaging with it (usually patients and families), lack of comfort using technology for medical visits and slow or poor internet connection [37, 40, 41, 49, 51, 56, 58]. Individuals with cognitive, vision or hearing impairments, limited decision-making capacity or who required an interpreter to speak with their care provider created challenges for virtual engagement [40, 41, 47, 53]. The lack of patient access to a smart phone or internet, as well as the providers' inability to read nonverbal cues and body language represented additional barriers to virtual care [40, 41, 48, 49, 51].

### Lessons learned through patient engagement

Despite many challenges to engaging with patients and families during COVID-19, several articles also reflected on lessons learned [33, 37, 39, 41–43, 49, 53–56, 58]. There was an identified need for more team-based models and approaches to virtual care and telehealth [37, 41, 43]. Authors explained that virtual care should reflect the care an individual would receive in person, so if a patient would be supported by a multidisciplinary team at an in-person clinic, they should receive the same level of support virtually [37]. The need for training prior to engaging in virtual care and telehealth was identified to ensure providers, patients and families were comfortable using the platform before participating in live sessions [41, 56]. Lastly, the responses to providing care differently (i.e. virtually) during COVID-19 should be used as an example for future disasters, that limit in-person care contact [53, 58]. Successful response strategies to maintain patient engagement during COVID-19 should be referred to and used as guidance if future disruptions were to occur [53].

## Discussion

To our knowledge, this scoping review was the first to examine what is known in the literature about patient engagement activities during the COVID-19 pandemic. Our findings showed a delineation between tools to support patients and caregivers in *receipt* of care interactions (most articles) and partnership activities in the *design* of care (few articles). Partnership activities like co-designing a new care delivery pathway or participating in decision making tables is how engagement is more traditionally defined. However, we took at broad view on engagement, in line with Carman's framework, to include direct care consultations, allowing us to capture the various ways in which health systems stayed connected to patients and families during the pandemic.

Based on Carman's continuum, the majority of published literature at the time of our scoping review was centered on activities at the level of direct care (e.g. consultations, care conferences, home monitoring, remote appointments/check-ups) [21]. In some cases, the focus of these consultations was not just on medical care needs, but on social determinants of health and ability to self-manage (perhaps due to heightened acknowledgement of social needs amidst the pandemic). Many organizations were incorporating virtual visits into their practice prior to the pandemic. Virtual care and telehealth were emerging in certain fields to target specific population groups, but these technologies were often resisted by organizations because of worries around privacy and security [62], with concerns about ownership of data, authorization of unspecified use and data security noted as reasons for the lack of large-scale uptake [63]. In spite of these concerns, it is possible that COVID-19 presented a window of opportunity for this work to be pushed forward and adopted at a rapid pace, as many Health Insurance Portability and Accountability Act (HIPAA) private health information confidentiality violations were waived during the pandemic to allow for the use of non-encrypted technologies for virtual care [64].

We also saw some examples of broader community engagement, also through use of technology, to not only share information, but to create a space for the community to connect, ask questions or share concerns [42, 55, 59]. Given the short time window of our scoping review, it is not surprising that individual engagement such as virtual consultations and home monitoring were implemented and published more quickly compared to other levels of engagement along Carman's continuum (e.g. partners in organizational governance and policy change) [21]. Changes to organization governance and policies often require more time, effort and resources. Overall, it is possible that the pandemic created the space, context and opportunity to quickly push forward and transition to a more virtual environment, which is reflected in the literature. A virtual environment is important for facilitating both individual interactions and patient partnership.

While it may seem that virtual care and the use of technology for providing health and social services do not reflect active patient engagement, such activities allow for capacity building and can incorporate models of shared decision-making, goal setting and patient autonomy in directing their own care. As such, these activities demonstrate a more active form of engagement and highlight how partnership may occur during virtual and direct care activities. However, it was not always clear in the included articles if these more active forms of direct care engagement were used.

Some initiatives did signify a deeper form of engagement by way of *bi-directional* information sharing (providing feedback to improve services and programs), as well as virtually connecting families to be part of clinical consultations due to visitor restrictions, similar to what we would see in activities such as bedside rounding [65]. Furthermore, and perhaps most promising, was the involvement of multiple stakeholders (patients, clients and patient

organizations) in a virtual environment in the design of a global patient experience survey for individuals with rheumatic disease, showing the potential of engagement during a crisis across all stages of the research cycle [34].

This scoping review highlighted the rapid transition to a virtual environment, through the use of online platforms, social media and telehealth. Almost all of the included articles used technology, in one form or another, to engage with patients and families during COVID-19, with virtual care being the most common (n = 13). Virtual care and the use of technology for patient engagement activities can open doors and increase opportunities for involvement [66, 67]. For example, accessibility, continuity of care, cost effectiveness, health outcomes, satisfaction and attention to equity (in terms of access and patients' social, cultural and health needs being addressed) were highlighted as potential benefits of virtual care in a rapid review conducted by Li and colleagues (2020) [66]. Similarly, a literature review of telehealth in rural Australia identified several benefits, including: lower costs, improved convenience accessing services and specialists and improved quality of services [67].

On the other hand, virtual care and technology use may also close doors and limit involvement from certain individuals and population groups in both clinical care interactions and, by extension, from patients and families involved in partnership activities (e.g., decision making tables, co-design activities). As identified in this review, engagement was impacted by poor internet connection, lack of technology or technology that did not support video conferencing and an individual's lack of knowledge or comfort using technology [37, 40, 41, 49, 51, 56, 58]. Many of these challenges with telehealth and virtual care have been echoed in the literature [66, 68–70]. Bandwidth issues and access to devices have been noted to negatively impact the ability to engage virtually, as well as an individual's overall experience [70, 71]. In addition to technology-related barriers to engagement, this review highlighted that individual level factors (i.e. health literacy, socioeconomic status) can also negatively impact virtual interactions, and as such, an individual's opportunity to engage in activities when they are technology-based [40, 41, 47, 53]. Social and economic factors have been noted to create challenges for virtual visits, as mental health issues and low levels of health literacy can impede conversations between healthcare professionals and patients [68]. Persons who are older, non-English speaking, unemployed, low income, live in rural areas or are a racial/ethnic minority (African American, Latino, Japanese, Chinese, Filipino, South Asian) can also experience a divide from virtual care [69]. Furthermore, the use of virtual care and technology has the potential to increase opportunities for involvement (for patients both in clinical care interactions and partnership activities), but it may also hinder them. It is critical to better understand how to ensure involvement is not limited or restricted when transitioning to online platforms. The "Person-Centred Guidelines for Preserving Family Presence in Challenging Times" report recognized that technology is not always an option for people and included examples of non-technology based ways to connect (through volunteers, medical students, etc.) who act as connectors between patients and families. This connector role could also serve as the interface between patients, caregivers and other stakeholders.

The challenges related to virtual care engagement (language, health literacy, socio-economic status) parallel the kinds of barriers noted in patient engagement activities generally [72, 73]. For example, Chegini and colleagues conducted a qualitative study in which low levels of health literacy, cultural barriers, ineffective patient education and patient unwillingness were identified as patient-related barriers impacting engagement in the safe delivery of care in hospital [72]. Since this scoping review pointed to equity challenges in engaging in virtual care it will be important to place emphasis on addressing these barriers as we move into the COVID-19 recovery period. Further, equity challenges in participating in patient engagement activities extend beyond virtual care and can occur along the continuum of engagement; thus

highlighting the importance of applying a health equity lens to engagement, which is reflexive, intersectional and trauma-informed [74].

In using Carman's framework as a guide, we identified several limitations that could be addressed in future work. Despite having three levels of engagement across a continuum of three categories, we identified challenges categorizing direct care level engagement activities. This was based on a limited ability to highlight differences in these activities. For example, the majority of patient engagement activities were categorized at the direct care level, as consultation or involvement. However, we identified a number of differences in these engagement activities that were lost with this categorization and we were able to create a more nuanced set of categories, many of which fit under the umbrella of direct are activities. Based on the purposes of this review, it was important to highlight these nuances and differences in the engagement activities, which is why we conducted a more in-depth analysis following the categorization according to the framework. Additionally, Carman et al.'s definition of patient engagement involves an active partnership between patients, families, representatives and health professionals; however, the levels of engagement along the framework continuum do not all include active partnership [21]. It is important for the definition and framework to present consistent information to eliminate confusion around the classification of activities as patient engagement. Overall, this framework served as a fundamental starting point from which we adapted to better reflect the nuances discovered in our scoping review.

## Future work

Based on the findings from this scoping review, several areas of future work have been identified. There is a need for more original research on all patient engagement activities (clinical care interactions and partnership activities) occurring during the pandemic to better describe engagement work from different perspectives, as well as exposure factors related to success. For example, research evaluating the use of technology for other types of patient engagement (i.e., advisory committees, planning and decision-making activities), exploring the perspectives of patients, families, providers and researchers regarding engagement activities during COVID-19 and gaining a better understanding of how organizations transitioned to maintain patient engagement. It is critical to understand strategies that have worked, or not worked, to allow organizations to continue patient engagement activities. The findings of this scoping review may imply that other types of engagement activities (e.g., partnering in organizational design and governance, policy decisions) were not occurring as much as individual level engagement, but it is highly likely that our results were more a reflection that this work, and the learning that can be gained from it, may not yet be published. However, it is important to explore this area further to identify if (and why) partnership activities at the organizational level came to a halt during the initial stages of the pandemic and beyond. Understanding the structures and processes that allow for engagement activities to continue during times of major disruption is critical for improving patient engagement activities and overall experiences. Environmental scans, original qualitative research and case studies, for example, are recommended methods for capturing these other, organizational level forms of engagement and is the next phase of work being conducted by our team. Additionally, as the focus of this scoping review was on patient engagement activities occurring during the first six months of the pandemic, it is important to explore if, and how, engagement approaches have evolved over the course of the pandemic. Lastly, there is a need to explore if, and how, patients, caregivers and families can remain involved in engagement activities without relying solely on technology.

## Limitations

There are a few limitations to be noted. First, based on the rapid publication of COVID-related research and topics, it is possible that relevant articles were missed because of the state of indexing. Second, our search was conducted in English, so it is possible that articles published in other languages were missed. Third, based on the current state of the literature, with many of the included articles being editorials and commentaries that lacked contextual information, we did not conduct a critical appraisal of the quality of the articles. However, this is not a requirement for scoping reviews [27]. Lastly, we acknowledge that this review is not reflective of all the patient engagement work that is being done during COVID-19, as there is the potential that engagement is occurring on different platforms or other levels of engagement that take longer to implement have not yet been written about or published. For example, while we included articles that used social media for engagement, we did not specifically search social media (Twitter, Facebook, Sina Weibo, etc.) for posts related to engagement. Despite these limitations, this review presents a summary of the work that was implemented and published during the first six months of the pandemic.

## Conclusions

This scoping review identified a number of examples of how healthcare systems stayed connected to patients and families during the pandemic. Though we had exclusively looked for examples of patient engagement activities, we found few examples of patient partnership and more examples of direct care consultations via technology as well as broader community engagement for purposes of sharing and receiving information related to the pandemic. Other research methods to explore specific contexts and initiatives (e.g., qualitative investigations or case studies) may better unpack the full spectrum of patient engagement activities that occurred during the pandemic and give greater insight into the barriers and facilitators of sustaining these activities.

## Supporting information

**S1 Table. MEDLINE search strategy.**
(DOCX)

**S2 Table. Preferred reporting items for systematic reviews and meta-analyses extension for scoping reviews (PRISMA-ScR) checklist.**
(DOCX)

## Acknowledgments

The authors would like to thank the librarian at Trillium Health Partners (Antonia Giannarakos) for her guidance on the search strategy. We would also like to thank Dr. Alison Freeland, Sandy Dayes, Shiza Sheikh and Aditi Desai from Trillium Health Partners and Dr. Carol Fancott, Julie Drury and Jessie Checkley from Healthcare Excellence Canada for their ongoing support throughout the project. The findings and reflections in this paper do not necessarily reflect the views of our Collaborators and Funders.

## Author Contributions

**Conceptualization:** Kerry Kuluski.

**Data curation:** Kerry Kuluski.

**Formal analysis:** Lauren Cadel, Michelle Marcinow, Jane Sandercock, Penny Dowedoff, Kerry Kuluski.

**Funding acquisition:** Kerry Kuluski.

**Investigation:** Michelle Marcinow, Kerry Kuluski.

**Methodology:** Kerry Kuluski.

**Project administration:** Lauren Cadel, Kerry Kuluski.

**Supervision:** Kerry Kuluski.

**Validation:** Sara J. T. Guilcher, Kerry Kuluski.

**Visualization:** Kerry Kuluski.

**Writing – original draft:** Lauren Cadel, Michelle Marcinow, Jane Sandercock, Penny Dowedoff, Kerry Kuluski.

**Writing – review & editing:** Lauren Cadel, Michelle Marcinow, Jane Sandercock, Penny Dowedoff, Sara J. T. Guilcher, Alies Maybee, Susan Law, Kerry Kuluski.

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
