## [Decision Letter · Decision Letter 0]

6 May 2021

PONE-D-20-38436

A scoping review of patient engagement activities during COVID-19: More consultation than partnership

PLOS ONE

Dear Dr. Kuluski,

Thank you for submitting your manuscript to PLOS ONE. After careful consideration, we feel that it has merit but does not fully meet PLOS ONE’s publication criteria as it currently stands. Therefore, we invite you to submit a revised version of the manuscript that addresses the points raised during the review process.

Please ensure that you efficiently respond to the reservations raised by the reviewers, especially the reviewer No 2. The issues raised are well defined and rather serious; without providing relevant amendments and extensions, I do not think you paper can be proceeded further.

We look forward to receiving your revised manuscript.

Kind regards,

Mariusz Duplaga, Ph.D., M.D., Ass. Prof.

Academic Editor

PLOS ONE

Journal Requirements:

3. In the manuscript title page, it is not clear with which organisation 'Patient Partner, Canada' is affiliated.

Please amend your list of authors in the manuscript to ensure that each author is linked to an affiliation. Authors’ affiliations should reflect the institution where the work was done (if authors moved subsequently, you can also list the new affiliation stating “current affiliation:….” as necessary).

Reviewers' comments:

Reviewer's Responses to Questions

**Comments to the Author**

1. Is the manuscript technically sound, and do the data support the conclusions?

Reviewer #1: Yes

Reviewer #2: No

2. Has the statistical analysis been performed appropriately and rigorously? 

Reviewer #1: Yes

Reviewer #2: N/A

3. Have the authors made all data underlying the findings in their manuscript fully available?

Reviewer #1: Yes

Reviewer #2: Yes

4. Is the manuscript presented in an intelligible fashion and written in standard English?

Reviewer #1: Yes

Reviewer #2: Yes

5. Review Comments to the Author

Reviewer #1: This study aimed to determine what has been done in terms of patient engagement activities during the COVID-19 pandemic. The authors collected over 700 articles, performed screening, and selected 29 suitable articles for content analysis. The study identified 4 major topics related to patient engagement, and found that the majority of patient engagement activities were direct care consultation rather than patient partnership.

Patient engagement is a promising area in the field of health care education. Having patients articulate their experiences and viewpoints helps those taking part in training to appreciate the patient perspective and the importance of preserving trust between clinicians and patients. Thus, studying patient engagement activities during this COVID-19 pandemic may provide valuable insights to medical practitioners, patients and even the government to better prepare for future crises.

The analyses were carefully performed and the manuscript was well-written. Only some minor issues to be addressed:

On page 13, line 185, "Four main categories were identified: (1) Engagement through Virtual Care; (2) Engagement through Other Technology; (3) Engagement for Service Improvements/Recommendations; and (4) Factors Impacting Patient Engagement." How were the four categories derived? Were there more than 4 categories that were considered before the "4 main" were selected? Some better explanations could have given to how these four were identified.

I understand each of the n represents an article, but for some of the n's, it was unclear whether they overlapped.

For example, on page 23, line 282-284, "Patient engagement activities through the use of other technology (e.g. social media, online-based interventions) were identified as a way of sharing information (n=4), creating connections (n=4) and gathering information (n=6)."

Did the n's came from different articles or the same articles? Since these subcategories are not mutually exclusive, does that mean the apparent 15 (4+4+6) articles of "Engagement through Other Technology" could have actually just came from 6 articles?

The same question applies to the n's for "Engagement through Virtual Care" section.

The result that there were more consultations than patient partnerships seems to be rather within expectations. It is also unclear to me what the significance of this means. If the author believe that it is an interesting result, please explain why. The author should also better justify why this is an important finding, and more clearly spell out the significance of it.

Reviewer #2: Thank you for the opportunity to review this manuscript which aimed to scope Covid-19 related patient engagement activities in health care during the early Covid pandemic period. Overall the manuscript was easy to follow, although I identified a number of conceptual and methodological gaps which I feel should be addressed to strengthen the project prior to publication.

Conceptually, I identified multiple issues:

1) the project is framed on the question of the importance of understanding, what, if any, patient engagement activities occurred during the pandemic. However, the authors have not sufficiently conveyed an argument about WHY this is an important question to answer. What are the implications of answering this question?

2) I’m also questioning to some extent the relevance of a review of peer-reviewed literature to search for Patient engagement in healthcare during covid-19, because the peer reviewed literature is academic in focus and I’m uncertain about the extent to which academic studies on delivery of healthcare would accurately represent the extent of patient engagement in care delivery. Even the grey literature search relied to some extent on a publication of some sort, and given the rapid evolution of the pandemic and strain on healthcare systems during this time, I’m not sure that published evidence is a reliable source to answer this particular question of interest.

3) The project is also framed around engagement of patients, families and caregivers in health care delivery and health care systems, and uses Carman et al’s definition and framework to guide this work. While I appreciated the attempt to use this framework as an overarching approach, I had a number of concerns with how it played out over the project, leading me to question its appropriateness for this analysis. First, by the authors’ admissions, components of the framework are at odds with the definition, whereby the stated definition of patient engagement emphasizes active partnership at various levels, but then goes on to also include passive forms of engagement within the framework as well. This proves problematic when the results of the scoping review are weighed so heavily towards the most passive types of engagement in covid-19 that I seriously question if we should be considering these as meaningful categories of engagement at all—to me virtual care delivery is simply that, and doesn’t in and of itself meet the bar for qualifying as engagement.

Methodologically, there are also several major issues:

1) The authors have not convinced me that they conducted an in-depth thematic analysis. Details of the methods are not at all described (one vague sentence only with no reference provided), and it seems to me to be closer to content analysis (debatable if conventional or directed content analysis). I recommend the authors add significantly more detail enhance the methodological coherence of the project.

2) The date range for analysis appears rather arbitrary, identified as “first stage of covid-19” and bounded between March 2020 and July 2020, without any substantive consideration of what that means. Also given that the search is almost a year old, this ties back to my earlier question of wanting to better justify why studying this particular period is important, given that we have arguably come a long way since then. Or if an update is possible, it would be perhaps of more use to compare if engagement approaches evolved over the pandemic.

Specific comments:

- Title emphases engagement activities during covid-19, but it would be more accurate to convey that its about PE activities about covid-19 as well, as it does not appear the review was looking for any engagement during the pandemic. This could be clarified throughout the manuscript.

- Basic covid-19 statistics in the opening paragraph should be updated as these are now out-of-date

- Please reference the statement on line 75-78: “At each level, there is a continuum of engagement from consultation, to involvement, to partnership and shared leadership. Each stage of the continuum involves increased participation and collaboration from those being engaged, as they progress from passive information sharing to active partnership in the development and evaluation of healthcare programs and policies”.

- Please reference the statement on line 88-89 that: “many patient and family advisory committees, at least in the Canadian context, were put on hold.”

- Within the research question, “ What is known in the literature about work that has been done internationally within healthcare, government and academic organizations on patient engagement during the COVID-19 pandemic?” I am unclear about the relevance about asking about patient engagement within academic organizations as this seems incongruent with the stated project focus on healthcare delivery, and ties back to some of my other identified issues.

- The results section reads less as a synthesis of findings and more as a narrative description of each of the included studies. This needs to be reworked.

- In light of my other comments, I do not follow all of the categories identified—and again, these feel like domain summaries, not themes. “Engagement through virtual care”- as noted I question delivery of care as engagement in the way it was presented in the introduction. The sub-headings and categories are not clearly explained so the organizational structure is difficult to follow. There are also cross-cutting themes that are not highlighted—such as the use of technology for connection—both through virtual care and beyond. Lessons learned are distinct from patient-reported barriers and facilitators, and I think this should be emphasized.

- In the discussion, the comment “Our findings showed a delineation between tools to support patients and caregivers in receipt of care interactions (most articles) and partnership activities in the design of care (few articles).” Is helpful, although engagement hasn’t been framed in this way at all to this point in the paper, this would be helpful to the reader much earlier in the introduction.

- In light of my comments, the discussion requires a significant revision. There is much focus on virtual care, but not much reflection or critical discussion about whether use of technology in itself constitutes engagement. Is this just merely a new mode of engagement? Virtual vs in person? Is the mode the critical aspect of engagement we are concerned with?

6. PLOS authors have the option to publish the peer review history of their article (what does this mean?). If published, this will include your full peer review and any attached files.

Reviewer #1: No

Reviewer #2: No

---

## [Author Response · Author response to Decision Letter 0]

2 Jul 2021

Please note: all references to page and line numbers correspond to the track changed version of the manuscript.

The manuscript meets all of the journal’s style requirements.

PLOS requires an ORCID iD for the corresponding author in Editorial Manager on papers submitted after December 6th, 2016. Please ensure that you have an ORCID iD and that it is validated in Editorial Manager. To do this, go to ‘Update my Information’ (in the upper left-hand corner of the main menu), and click on the Fetch/Validate link next to the ORCID field. This will take you to the ORCID site and allow you to create a new iD or authenticate a pre-existing iD in Editorial Manager. Please see the following video for instructions on linking an ORCID iD to your Editorial Manager account: https://www.youtube.com/watch?v=_xcclfuvtxQ

The corresponding author has an ORCID ID and it is validated in Editorial Manager.

In the manuscript title page, it is not clear with which organisation 'Patient Partner, Canada' is affiliated.

Please amend your list of authors in the manuscript to ensure that each author is linked to an affiliation. Authors’ affiliations should reflect the institution where the work was done (if authors moved subsequently, you can also list the new affiliation stating “current affiliation:….” as necessary).

The author is a patient partner who contributed to both the project and manuscript and is not affiliated with a specific institution. 

Reviewer 1:

This study aimed to determine what has been done in terms of patient engagement activities during the COVID-19 pandemic. The authors collected over 700 articles, performed screening, and selected 29 suitable articles for content analysis. The study identified 4 major topics related to patient engagement, and found that the majority of patient engagement activities were direct care consultation rather than patient partnership.

Patient engagement is a promising area in the field of health care education. Having patients articulate their experiences and viewpoints helps those taking part in training to appreciate the patient perspective and the importance of preserving trust between clinicians and patients. Thus, studying patient engagement activities during this COVID-19 pandemic may provide valuable insights to medical practitioners, patients and even the government to better prepare for future crises.

The analyses were carefully performed and the manuscript was well-written. Only some minor issues to be addressed:

Thank you for this comment.

On page 13, line 185, "Four main categories were identified: (1) Engagement through Virtual Care; (2) Engagement through Other Technology; (3) Engagement for Service Improvements/Recommendations; and (4) Factors Impacting Patient Engagement." How were the four categories derived? Were there more than 4 categories that were considered before the "4 main" were selected? Some better explanations could have given to how these four were identified.

We have added additional information to better explain how the four categories were developed. See pages 10-11, lines 215-222. 

I understand each of the n represents an article, but for some of the n's, it was unclear whether they overlapped.

For example, on page 23, line 282-284, "Patient engagement activities through the use of other technology (e.g. social media, online-based interventions) were identified as a way of sharing information (n=4), creating connections (n=4) and gathering information (n=6)."

Did the n's came from different articles or the same articles? Since these subcategories are not mutually exclusive, does that mean the apparent 15 (4+4+6) articles of "Engagement through Other Technology" could have actually just came from 6 articles?

There was very minimal crossover of articles between subcategories, it was more common for there to be crossover in the main categories (e.g. virtual care and service improvement). We have added the references in the opening paragraphs for each of the categories/sub categories to help clarify this and show where any crossover occurred. See page 26, lines 358-360.

The same question applies to the n's for "Engagement through Virtual Care" section.

There was very minimal crossover of articles between subcategories, it was more common for there to be crossover in the main categories (e.g. virtual care and service improvement). We have added the references in the opening paragraphs for each of the categories/sub categories to help clarify this and show where any crossover occurred. See page 22, lines 250-253.

The result that there were more consultations than patient partnerships seems to be rather [unclear?] within expectations. It is also unclear to me what the significance of this means. If the author believe that it is an interesting result, please explain why. The author should also better justify why this is an important finding, and more clearly spell out the significance of it.

The result that there were more consultations than patient partnerships was not surprising to us, given the focus of the review was on activities during the first six months after the pandemic was declared. We believe that the pandemic created a platform that accelerated the focus and work around virtual care (see discussion page 33, line 519). This allowed patients and families to stay connected to providers, as well as the healthcare system more broadly; however, with the focus on virtual consultations, there were minimal examples of other forms of engagement along the continuum (partnership and shared leadership). The lack of engagement activities in these areas represents an area of future work to further explore. We have added this information to future work, see pages 39, lines 646-651.

Reviewer 2:

Thank you for the opportunity to review this manuscript which aimed to scope Covid-19 related patient engagement activities in health care during the early Covid pandemic period. Overall the manuscript was easy to follow, although I identified a number of conceptual and methodological gaps which I feel should be addressed to strengthen the project prior to publication.

Thank you for your comments. 

Conceptually, I identified multiple issues:

1) the project is framed on the question of the importance of understanding, what, if any, patient engagement activities occurred during the pandemic. However, the authors have not sufficiently conveyed an argument about WHY this is an important question to answer. What are the implications of answering this question?

Thank you for this comment, we have significantly revised the introduction to better convey why it is important to understand patient engagement activities occurring during the pandemic. See pages 6-7, lines 127-133.

2) I’m also questioning to some extent the relevance of a review of peer-reviewed literature to search for Patient engagement in healthcare during covid-19, because the peer reviewed literature is academic in focus and I’m uncertain about the extent to which academic studies on delivery of healthcare would accurately represent the extent of patient engagement in care delivery. Even the grey literature search relied to some extent on a publication of some sort, and given the rapid evolution of the pandemic and strain on healthcare systems during this time, I’m not sure that published evidence is a reliable source to answer this particular question of interest.

We do acknowledge that a scoping review may not capture all of the patient engagement activities occurring during the first six months of the pandemic (noted in the limitations section); however, this approach is one way to synthesize evidence and was appropriate for identifying the extent and types of available evidence pertaining to this topic. By not limiting the review to peer-reviewed literature, we were able to identify additional patient engagement activities through the grey literature searches. A scoping review approach also allowed for us to scan international literature. 

3) The project is also framed around engagement of patients, families and caregivers in health care delivery and health care systems, and uses Carman et al’s definition and framework to guide this work. While I appreciated the attempt to use this framework as an overarching approach, I had a number of concerns with how it played out over the project, leading me to question its appropriateness for this analysis. First, by the authors’ admissions, components of the framework are at odds with the definition, whereby the stated definition of patient engagement emphasizes active partnership at various levels, but then goes on to also include passive forms of engagement within the framework as well. This proves problematic when the results of the scoping review are weighed so heavily towards the most passive types of engagement in covid-19 that I seriously question if we should be considering these as meaningful categories of engagement at all—to me virtual care delivery is simply that, and doesn’t in and of itself meet the bar for qualifying as engagement.

This framework allowed us to identify both active and passive types of patient engagement. In doing so, we were able to categorize a variety of activities that were occurring to keep patients and families connected to the healthcare system. While we acknowledge that there was a disconnect between the definition (active) and the framework (active and passive), we think it was important to highlight all types of activities occurring along the continuum, in order to identify current gaps and where more work is needed. Our results weighed more heavily towards the more passive types of engagement because that was the state of the literature; however, it is important to note that other examples of more “authentic engagement” have also been highlighted. By using this framework, we were able to add to, and further, our conceptual understanding of patient engagement.

Methodologically, there are also several major issues:

1) The authors have not convinced me that they conducted an in-depth thematic analysis. Details of the methods are not at all described (one vague sentence only with no reference provided), and it seems to me to be closer to content analysis (debatable if conventional or directed content analysis). I recommend the authors add significantly more detail enhance the methodological coherence of the project.

We appreciate this comment and agree that our analysis approach aligns with conventional content analysis. We have added additional information and a reference about the analysis process in order to enhance the methodological coherence. See page 10, lines 213-220. 

2) The date range for analysis appears rather arbitrary, identified as “first stage of covid-19” and bounded between March 2020 and July 2020, without any substantive consideration of what that means. Also given that the search is almost a year old, this ties back to my earlier question of wanting to better justify why studying this particular period is important, given that we have arguably come a long way since then. Or if an update is possible, it would be perhaps of more use to compare if engagement approaches evolved over the pandemic.

We appreciate this comment and have added information to better describe the timeframe in which this review is bounded (see page 9, lines 179-181). We have focused this review on the first 6 months after the pandemic was declared in order to gain a deeper understanding of the initial shift of activities. We agree with the comment that we have come a long way since the beginning of the pandemic and do think it is important to see how patient engagement has evolved over this time; however, since that is beyond the scope of this review, we have noted it as an area of future work (see pages 39-40, lines 653-656). We are also in the process of conducting case studies that will further explore patient engagement activities during the pandemic and how they have evolved over time.

Specific comments:

- Title emphases engagement activities during covid-19, but it would be more accurate to convey that its about PE activities about covid-19 as well, as it does not appear the review was looking for any engagement during the pandemic. This could be clarified throughout the manuscript.

We did not limit engagement activities to activities about COVID-19, despite the majority being specific to the pandemic. We have added some information to the methods and to the results to clarify that engagement activities did not have to be specific to COVID-19 for inclusion. See page 9, lines 177-179 and page 15, lines 244-245.

- Basic covid-19 statistics in the opening paragraph should be updated as these are now out-of-date

The COVID-19 statistics in the opening paragraph have been updated. See page 4, line 62-63. 

- Please reference the statement on line 75-78: “At each level, there is a continuum of engagement from consultation, to involvement, to partnership and shared leadership. Each stage of the continuum involves increased participation and collaboration from those being engaged, as they progress from passive information sharing to active partnership in the development and evaluation of healthcare programs and policies”.

We have added a reference for this statement.

- Please reference the statement on line 88-89 that: “many patient and family advisory committees, at least in the Canadian context, were put on hold.”

We have added a reference for this statement.

- Within the research question, “What is known in the literature about work that has been done internationally within healthcare, government and academic organizations on patient engagement during the COVID-19 pandemic?” I am unclear about the relevance about asking about patient engagement within academic organizations as this seems incongruent with the stated project focus on healthcare delivery, and ties back to some of my other identified issues.

We had included academic organizations in the research question to ensure university-affiliated activities (within the healthcare context) were not excluded; however, we do understand how this may create confusion, so we have removed academic organizations from the research question (as the focus was on healthcare delivery and staying connected to the healthcare system).

- The results section reads less as a synthesis of findings and more as a narrative description of each of the included studies. This needs to be reworked.

Thank you for this comment; we have significantly revised the results to be more of a synthesis and reduce the narrative descriptions of the included studies. See pages 22-32.

- In light of my other comments, I do not follow all of the categories identified—and again, these feel like domain summaries, not themes. “Engagement through virtual care”- as noted I question delivery of care as engagement in the way it was presented in the introduction. The sub-headings and categories are not clearly explained so the organizational structure is difficult to follow. There are also cross-cutting themes that are not highlighted—such as the use of technology for connection—both through virtual care and beyond. Lessons learned are distinct from patient-reported barriers and facilitators, and I think this should be emphasized.

We have revised the analysis section of the methods to describe the process as a content analysis and thus, the sections of the results are frame as categories, rather than themes. We have also created a figure to help display the categories and sub-categories. Cross-cutting examples are highlighted in the Virtual Care for Clinical Service category (Tenforde and Annis references). We have added a line to explicitly state this. We have added a category to separate lessons learned from barriers and facilitators (see page 32, line 486).

- In the discussion, the comment “Our findings showed a delineation between tools to support patients and caregivers in receipt of care interactions (most articles) and partnership activities in the design of care (few articles).” Is helpful, although engagement hasn’t been framed in this way at all to this point in the paper, this would be helpful to the reader much earlier in the introduction.

We have added some detail in the introduction in order to better frame engagement and help orient readers that the review will include both active and passive patient engagement activities. See page 5, lines 88-106.

- In light of my comments, the discussion requires a significant revision. There is much focus on virtual care, but not much reflection or critical discussion about whether use of technology in itself constitutes engagement. Is this just merely a new mode of engagement? Virtual vs in person? Is the mode the critical aspect of engagement we are concerned with?

We appreciate this comment and have revised the discussion to include a paragraph that is more critical about what constitutes patient engagement. See page 34, lines 535-541.

---

## [Decision Letter · Decision Letter 1]

28 Jul 2021

PONE-D-20-38436R1

A scoping review of patient engagement activities during COVID-19: More consultation, less partnership

PLOS ONE

Dear Dr. Kuluski,

Thank you for submitting your manuscript to PLOS ONE. After careful consideration, we feel that it has merit but does not fully meet PLOS ONE’s publication criteria as it currently stands. Therefore, we invite you to submit a revised version of the manuscript that addresses the points raised during the review process.

Please consider the comments of the Reviewer 2 and try implement suggested amendments.

We look forward to receiving your revised manuscript.

Kind regards,

Mariusz Duplaga, Ph.D., M.D., Ass. Prof.

Academic Editor

PLOS ONE

Journal Requirements:

Reviewers' comments:

Reviewer's Responses to Questions

**Comments to the Author**

1. If the authors have adequately addressed your comments raised in a previous round of review and you feel that this manuscript is now acceptable for publication, you may indicate that here to bypass the “Comments to the Author” section, enter your conflict of interest statement in the “Confidential to Editor” section, and submit your "Accept" recommendation.

Reviewer #1: All comments have been addressed

Reviewer #2: (No Response)

2. Is the manuscript technically sound, and do the data support the conclusions?

Reviewer #1: Yes

Reviewer #2: Yes

3. Has the statistical analysis been performed appropriately and rigorously? 

Reviewer #1: Yes

Reviewer #2: N/A

4. Have the authors made all data underlying the findings in their manuscript fully available?

Reviewer #1: Yes

Reviewer #2: Yes

5. Is the manuscript presented in an intelligible fashion and written in standard English?

Reviewer #1: Yes

Reviewer #2: Yes

6. Review Comments to the Author

Reviewer #1: (No Response)

Reviewer #2: Thank you to the authors for reflecting on my feedback and revising the manuscript. I have only a few outstanding comments.

On further reflection, I realized that nowhere in the manuscript is the concept of engagement defined. Adding a definition in the introduction would increase clarity for readers. Also, could the authors please explain in the methods how engagement was determined—did papers self-identify as engagement and use the terminology, or did the reviewers identify engagement based on a comparison of their definition to reported descriptions in the included manuscripts? These points have implications for the interpretation of the findings, particularly in to the more superficial types of engagement in patient care noted by the authors.

I struggled with the statements in the study selection section around context. given publishing timelines, would it not be more relevant to clarify if the engagement activities occurred during the pandemic? Presumably would many studies published in early 2020 actually be reporting on issues/ activities from before the pandemic? I’m unclear how this was determined.

The organization and headings in table 2 are confusing to me. “Challenges” and “Lessons Learned” are not characteristics of strategies. Also, “Engagement category” column is reporting sub-categories. All told this table doesn’t align with the text reporting of results and is difficult to follow.

The revisions to the reporting of the findings through categories was generally helpful. However on further reflection, I question whether the sub-category “technology for sharing information” should be categorized as engagement, as this really seems to me to better align with classic definitions of dissemination of information and one way transmission of communication. Based on the descriptions provided, there is no evidence of any interaction, therefore I’m not sure it should be called engagement. This is also why adding definitions to clarify what the authors mean by engagement would be helpful here.

The statement on line 78-81, “While these examples refer to more ‘active’ forms of engagement, activities can be more broad in nature, ranging -- from clinical consultations to partnership activities (advisory groups and participation in policy decision making)” is important and requires a reference.

There are some incorrect references that need to be cleaned up (e.g. 21).

7. PLOS authors have the option to publish the peer review history of their article (what does this mean?). If published, this will include your full peer review and any attached files.

Reviewer #1: No

Reviewer #2: No

---

## [Author Response · Author response to Decision Letter 1]

29 Jul 2021

Journal Requirements:

Response: We have reviewed the reference list to ensure that it is complete and correct.

Reviewer #1: No response required

Reviewer #2: Thank you to the authors for reflecting on my feedback and revising the manuscript. I have only a few outstanding comments.

On further reflection, I realized that nowhere in the manuscript is the concept of engagement defined. Adding a definition in the introduction would increase clarity for readers. Also, could the authors please explain in the methods how engagement was determined—did papers self-identify as engagement and use the terminology, or did the reviewers identify engagement based on a comparison of their definition to reported descriptions in the included manuscripts? These points have implications for the interpretation of the findings, particularly in to the more superficial types of engagement in patient care noted by the authors.

Response: Thank you for noting this. Engagement was described, but we have re-integrated the definition in the introduction to increase clarity for the readers. Please refer to the Study Selection section of the methods for the explanation of how engagement was determined. The described activity had to align with at least one of the three core levels of engagement (along the continuum) as outlined in Carman and colleagues’ framework for patient and family engagement.

I struggled with the statements in the study selection section around context. given publishing timelines, would it not be more relevant to clarify if the engagement activities occurred during the pandemic? Presumably would many studies published in early 2020 actually be reporting on issues/ activities from before the pandemic? I’m unclear how this was determined.

Response: Please refer to our first inclusion criteria in the Study Selection section of the methods. Articles were required to be specific to COVID-19. So, while it is possible for studies published in early 2020 to be reporting on activities from before the pandemic, for this review, we required the articles to be specific to the context of the pandemic. If articles did not clearly state that the engagement activity was occurring during the pandemic, then it was excluded. 

The organization and headings in table 2 are confusing to me. “Challenges” and “Lessons Learned” are not characteristics of strategies. Also, “Engagement category” column is reporting sub-categories. All told this table doesn’t align with the text reporting of results and is difficult to follow.

Response: We have changed the title of Table 2 to ‘Characteristics of patient engagement activities’ to reduce potential confusion around the challenges and lessons learned columns. These columns are specific to the context of the activities and provide important contextual information. We have revised the “Engagement Category” column heading to reflect what is reported. We have reorganized the table to better align with the text presentation of the results. 

The revisions to the reporting of the findings through categories was generally helpful. However on further reflection, I question whether the sub-category “technology for sharing information” should be categorized as engagement, as this really seems to me to better align with classic definitions of dissemination of information and one way transmission of communication. Based on the descriptions provided, there is no evidence of any interaction, therefore I’m not sure it should be called engagement. This is also why adding definitions to clarify what the authors mean by engagement would be helpful here.

Response: We appreciate this comment and acknowledge that the use of technology for sharing information may be more of a one way transmission of communication. However, based on Carman and colleagues’ framework for patient and family engagement that was used to categorize engagement activities in this scoping review, these types of activities occur along the continuum of engagement and thus, we chose to include them. Please refer to the introduction for the definition of patient engagement, as well as an explanation of the framework used for conceptualizing and categorizing the engagement activities.

The statement on line 78-81, “While these examples refer to more ‘active’ forms of engagement, activities can be more broad in nature, ranging -- from clinical consultations to partnership activities (advisory groups and participation in policy decision making)” is important and requires a reference.

Response: We have added a reference to this statement.

There are some incorrect references that need to be cleaned up (e.g. 21).

Response: We have corrected the one invalid reference.

---

## [Editor Report · Decision Letter 2]

14 Sep 2021

A scoping review of patient engagement activities during COVID-19: More consultation, less partnership

PONE-D-20-38436R2

Dear Dr. Kuluski,

We’re pleased to inform you that your manuscript has been judged scientifically suitable for publication and will be formally accepted for publication once it meets all outstanding technical requirements.

Kind regards,

Lucinda Shen, MSc

Staff Editor

on behalf of 

Mariusz Duplaga, Ph.D, M.D

Additional Editor

PLOS ONE
---

## [Editor Report · Acceptance letter]

20 Sep 2021

PONE-D-20-38436R2 

A Scoping Review of Patient Engagement Activities during COVID-19: More Consultation, Less Partnership 

Dear Dr. Kuluski:

I'm pleased to inform you that your manuscript has been deemed suitable for publication in PLOS ONE. Congratulations! Your manuscript is now with our production department. 

Kind regards, 

on behalf of

Dr. Mariusz Duplaga 

Academic Editor

PLOS ONE